# Numerical Simulation on Air-Liquid Transient Flow and Regression Model on Air-Liquid Ratio of Air Induction Nozzle

Changxi Liu [1,2,3], Jun Hu [1,2,3,*], Yufei Li [1,2,3], Shengxue Zhao [1,2,3], Qingda Li [1,2,3] and Wei Zhang [1,2,3]

1   College of Engineering, Heilongjiang Bayi Agricultural University, Daqing 163319, China
2   Heilongjiang Province Conservation Tillage Engineering Technology Research Center, Daqing 163319, China
3   Key Laboratory of Soybean Mechanized Production, Ministry of Agriculture and Rural Affairs, Daqing 163319, China
*   Correspondence: gcxykj@126.com; Tel.: +86-138-3696-2331

**Abstract:** Air induction nozzle (AIN) has a special Venturi structure that has been widely used in the field of reducing the probability of drift of pesticide droplets and realizing precise application. The present research mainly adopts the method of comparative test and analyzes the difference between AIN and standard fan nozzle. However, the research on internal flow characteristics and air–liquid ratio (ALR) of AIN is very limited. In order to detect the air-liquid transient flow distribution and the influence of the geometric parameter structure of Venturi on the air–liquid ratio in the air induction nozzle, numerical simulation and air-liquid ratio prediction model of AIN combined with TD (Turbo Drop series) type Venturi tubes and ST110 (standard nozzle series) type fan nozzles are used. Based on the VOF (volume of fluid) model and Realizable k-ε turbulence control method, the TD-ST combined AIN is simulated numerically using open input and exit boundary conditions. The results show that the transient flow characteristic of the combined AIN is determined by the geometric structure of the Venturi tube, and the internal velocity and pressure change significantly at the Venturi angle. Under the same ST110 fan nozzle, the size of the larger TD Venturi tube will decrease the air phase content in the air–liquid flow. TD03-ST06 combined AIN has a maximum volume flow of 0.0092 (L/min) under 0.6 MPa. The air–liquid ratio regression model is established by designing the intake volume measurement system. According to this model, the influence law of tube size and spray parameters on the air–liquid ratio can be clarified. After variance analysis, it is proved that this model is suitable for air–liquid ratio prediction of TD-ST combined AIN. This study clarifies the air–liquid coupling law inside AIN and provides some reference for the quantitative analysis of the relationship between the geometric parameters, spray parameters, and the air–liquid ratio.

**Keywords:** air induction nozzle; VOF model; TD-ST combined AIN; air–liquid ratio; model prediction





## 1. Introduction

China is a major agricultural country with frequent and recurring pests [1]. The application of chemical pesticides to control disease, insect, and grass is the most effective control method at present [2,3]. The method of pesticide application has evolved over the years and gradually comes down to liquid agent spraying [4]. Large amounts of pesticides are lost in surrounding areas due to spray drift that result in health risks and environmental contamination. To improve drift reduction efforts in the crop protection field, researchers have been developing new spraying technologies to maximize the adsorption effect and increase the efficacy of crop protection products [5,6]. At present, there are many ways to reduce the drift of droplets, such as using cover spraying technology, cycle spraying technology, airflow-assisted spraying technology, and variable spraying technology [7–10]. The new plant protection equipment developed based on the aforementioned anti-drifting and high-efficiency application technology provides important assurance for pesticide reduction and also promotes the improvement of pesticide utilization rate and the implementation of pesticide reduction plan.

Compared with the above research methods, the most direct and effective way to solve droplet drift is to improve the anti-drift performance of the atomizing component, namely the nozzle [11]. The use of air induction nozzle (AIN) is an important technical method to reduce the probability of drift of pesticide aerosol and achieve precise application. AIN has a special Venturi structure compared to ordinary nozzles. When AIN is applied, high-speed flow induces air into the liquid phase to form microbubbles in the air–liquid mixed cavity, so as to expand the overall average size of the distribution of droplets to resist drift [12–14] . The interaction between air and liquid in AIN is of the greatest concern compared to conventional nozzle. The ratio of induced air to liquid is called air–liquid ratio (ALR), which is an important measure of atomization characteristics [15–17]. Previous scholars explored the AIN only from the intuitive manifestation, namely the increase of droplet size. Most scholars ignore the fundamental monitoring of air content. In addition, the irregular concave shape of the intake structure of most AIN makes it difficult to measure the air intake, resulting in little research on ALR.

At present, most AIN research focuses on the comparison with standard nozzle atomization, drift, and deposition properties. Guler [18] studied the particle width, coverage, and potential drift probability of a standard fan nozzle and AIN under the same model. Tang et al. [19] investigated the atomization characteristics of a standard fan nozzle and AIN under high-speed wind tunnel conditions. The test results can provide a reference for the selection of nozzle for fixed-wing manned aerospace application. Because of the small volume size of conventional AIN nozzles (such as IDK manufactured by LECHLER), it is difficult to improve the structure of the nozzles by exploring the internal flow characteristics of AIN. Therefore, the geometric structure dimension change of AIN is less studied to investigate the effects of its self-aspirating properties on atomization properties. The aperture of a conventional AIN nozzle is slightly larger than that of a standard fan nozzle of the same type. Therefore, some scholars believe that the increase in the size of AIN is due to the increase in its nozzle aperture. A variable aperture diameter and tube size AIN were designed and developed by Butler et al. [12], and its atomization characteristics are characterized macroscopically from the distribution and flow velocity of droplets. Vashahi et al. [20] made an air induction nozzle that is easy to replace and analyzed the influence of parameters such as the Venturi tube and aperture diameter on the AIN self-suction effect. Based on the previous research results and using STAR CCM + software (Version 10.06.010, from Siemens Company in Germany), Vashahi et al. [21] found that the fluid volume method (VOF) can be used to simulate the AIN of variable geometry structure. Although the self-made AIN can change the key geometric parameters of Venturi structure and explore its atomization characteristics, the geometric size of its nozzle differs greatly from that of the standard fan nozzle, and it is not suitable for the conclusion of a universal study.

Based on the above research background, a numerical simulation technique based on air–liquid coupling combined with experimental validation is used to study the transient flow characteristics and air–liquid ratio regression model of combination AIN. This study provides a certain design basis for the production of AIN in the future and also provides a reference for popularizing AIN for anti-drift spray operations.

## 2. Materials and Methods

### 2.1. Test Materials

The integrated production AIN has fixed the Venturi tube diameter and the nozzle model, making it difficult to change the single variable for testing. In addition, the volume of the integrated AIN nozzle is small and the structure of the air inlet is complex, so it is difficult to measure and analyze its self-suction characteristics. In this paper, Turbo Drop (TD) series Venturi tube produced by Agrotop Company in Germany and ST110 series fan nozzle produced by Lechler Company in Germany (Figure 1) were combined as test AIN (Hu et al. [22]). The aperture ratio of TD02, TD03, and TD04 are 0.69, 0.76, 0.85,

respectively. The slit width of ST-04, ST-05, ST-06 and ST-08 are 0.99 mm, 1.2 mm, 1.4 mm, 1.8 mm, respectively.

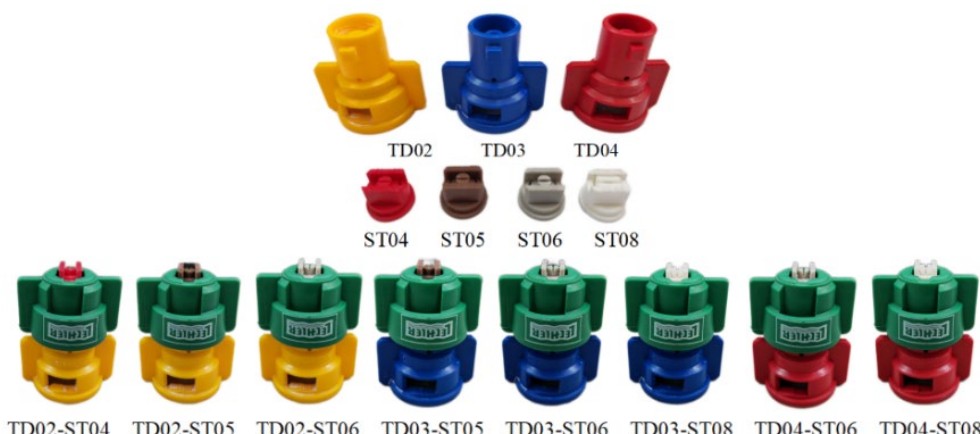

**Figure 1.** Test combination of TD-ST type AIN.

The TD-ST combined AIN works on the same principle as the conventional integrated AIN. It is based on the Venturi effect generated by the high-speed flow. The air is drawn from the inlet into the nozzle so that the sprayed droplets have a microbubble structure and resist drift (Figure 2).

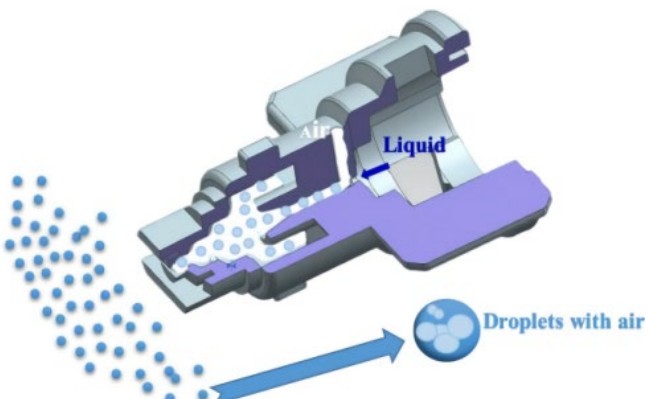

**Figure 2.** TD-ST combined AIN operating principle.

### 2.2. Test Equipment and Method

The test of TD-ST combination AIN air-to-liquid ratio (ALR) is carried out at HARDI Plant Protection Mechanics College, Heilongjiang Bayi Agricultural University. As shown in Figure 3, the test liquid is supplied by an air compressor pressurizing the reservoir. The liquid pressure is monitored by the Chinese Suxun SUX 90A-0-10 MPa diffusion silicon pressure sensor. The liquid receiver is set at the bottom of the nozzle to collect the liquid in real time.

As shown in Figure 4, the Siagro (Santa Clara, CA, USA) MF 4700-50-R-BV-A gas mass flow sensor is placed at the front of the air transport channel where the Venturi tube is connected to the gas transmission seal mold to measure air flow. The sensor can monitor a flow range of 0.0001~10 L/min. The communication output is RS485 modbus RTU, the communication interface is R1/4, and the test accuracy is $\pm (1.5 + 0.2FS)$%. To prevent the test gas from escaping, the Venturi tube is sealed with a special rubber mold between the nozzle fastening mold. When the spray nozzle is sprayed, the external air is drawn into the gas transmission channel from the Venturi tube and collected by the gas mass flow sensor. At this point, the electrical signal module is converted to data and stored in the Siargo User Application gas mass flow recording software.

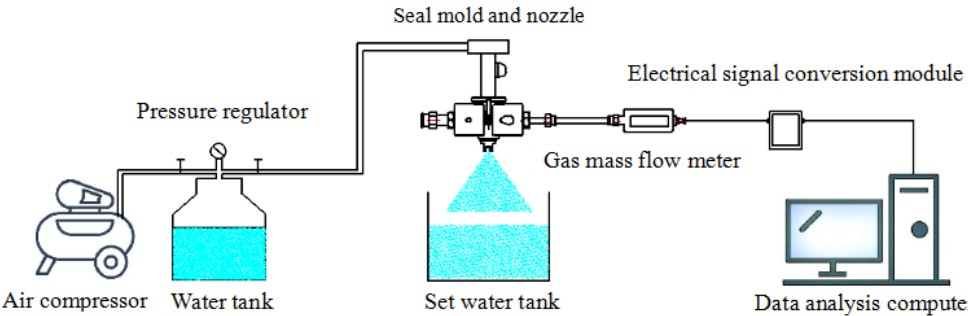

**Figure 3.** ALR measurement test.

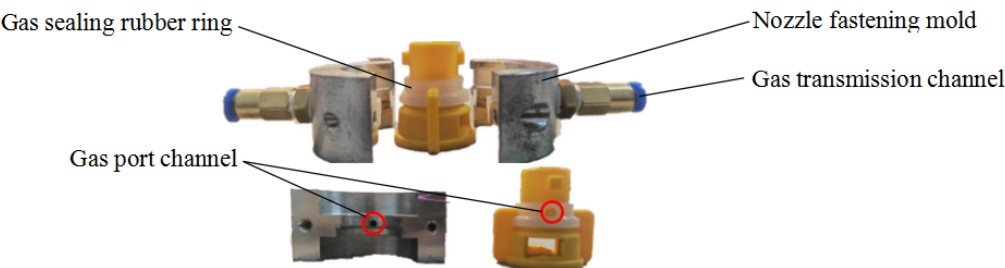

**Figure 4.** Venturi tube and gas transmission fastening mold installation.

*2.3. Numerical Simulation*

2.3.1. Theoretical Analysis

AIN is an atomized nozzle that uses jet technology to mix air and water to form a two-phase flow whose working principle is similar to that of a jet pump. Through a special Venturi structure, the flow rate increases rapidly during internal contraction, and after the liquid is ejected from the compression segment, a vacuum zone is formed near the exit, and the outside air is inhaled. The liquid and air enter the air–liquid mixed compression section, where the two phases are mixed, the diffusion section is joined together, an air–liquid two-phase flow is formed in the diffusion cavity, and then a large droplet filled with air bubbles is sprayed through the nozzle.

According to the air–liquid fluid dynamics, the key to great AIN atomization is to form a stable bubbly two-phase flow in the diffusion cavity. The steady flow state of the bubble air–liquid flow is related to the diameter and length of the mixing chamber and to the properties of the liquid. For the study of air–liquid two-phase flow in AIN, it is approximated that when the flow is uniform, the critical velocity of bubble-like flow is as follows:

$$c = \sqrt{\frac{\rho_a}{\beta^2 \rho_a + \beta(1-\beta)\rho_l}} c_a \tag{1}$$

The average density of air–liquid two-phase flow is:

$$\rho = \frac{1 + ALR}{\frac{1}{\rho_l} + \frac{ALR}{\rho_a}} \tag{2}$$

The critical flow of the nozzle is:

$$M = \mu A c \rho = \mu A c_a \rho_a \sqrt{\frac{1 + ALR}{ALR}} \tag{3}$$

where *ALR* is the ratio of air to liquid; *M* is the critical flow (L·min$^{-1}$); $\rho_a$ is air density (kg·m$^{-3}$); $\rho_l$ is liquid density (kg·m$^{-3}$); $c_a$ is the airflow velocity (m·s$^{-1}$); $\beta$ is the void fraction coefficient; *A* is the shrinkage cavity area (mm$^2$); and $\mu$ is the flow coefficient.

The flow coefficient is usually 0.58~0.65. When the spraying liquid pressure is determined, the pre-orifice $dp$ can be calculated as follows:

$$dp = \sqrt{\frac{M_l}{\frac{\pi}{4}\mu[2\rho_l(P_l - P_m)]^{0.5}}} \tag{4}$$

where $P_m$ is the mixing chamber pressure (Pa) and $P_l$ is liquid pressure (Pa).

The relationship of ALR is as follows:

$$ALR = \frac{\rho_a}{\rho_l}\frac{\beta}{1-\beta} = \frac{M_a}{M_l} \tag{5}$$

where $Ma$ is the air mass flow rate (L·min$^{-1}$) and $M_l$ is the liquid mass flow rate (L·min$^{-1}$).

The simulation model of TD-ST combinatorial AIN key parameters is shown in Figure 5. As can be seen from (1)~(5), the effect of $dp$ on liquid mass flow characteristics and operating parameters is equally significant. According to Vashahi et al. [21], the $da$ (air inlet diameter) increases the gas mass flow rate Ma, and $Pm$ is also an important reason for stabilization, mainly determined by $\theta$ (Venturi angle) and $dp/dt$ (pipe diameter ratio). Therefore, $dp$, $\theta$, $dt$, and $da$ are key parameters for AIN stabilization.

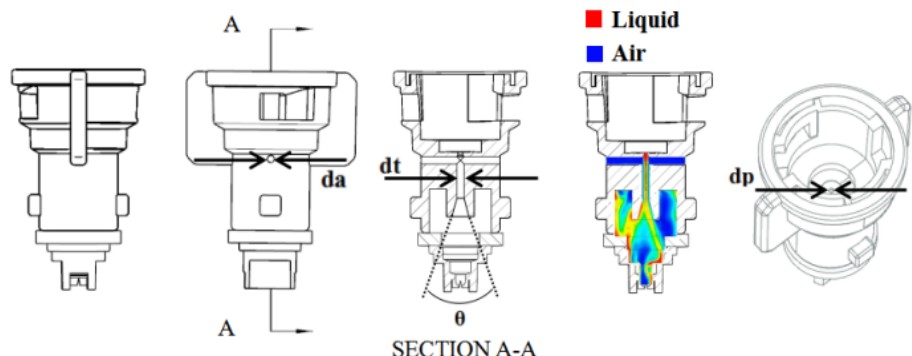

**Figure 5.** Simulation model of TD-ST type AIN with key geometric ruler. Note: The controlled design parameters are air inlet diameter (*da*), liquid inlet throat diameter (*dt*), pre-orifice diameter (*dp*), and Venturi angle (*θ*).

### 2.3.2. CFD Numerical Simulation Method

(1)   Model building

As shown in Figure 6, TD-ST type AIN is mapped with reverse engineering techniques and modeled with UG NX10.0 (computer modeling software). In order to analyze the internal flow characteristics and air–liquid ratio of TD-ST type AIN under different geometric parameters, the computational fluid dynamics method of CFD (ANSYS 19.0) software is used for numerical simulation.

(2)   Computational domain construction

The flow field computing domain of AIN after model simplification is established in CFX grid software. The flow field is divided into two parts: an internal AIN air–liquid two-phase calculation domain, as shown in Figure 7a, and an external AIN nozzle atomization calculation domain, as shown in Figure 7b.

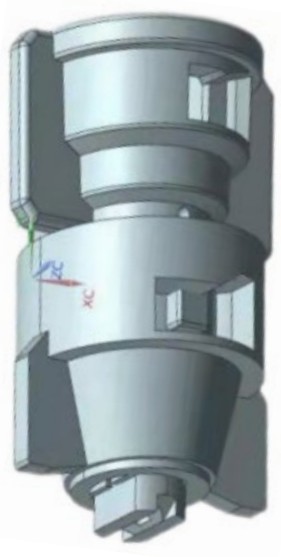

**Figure 6.** TD-ST type AIN model.

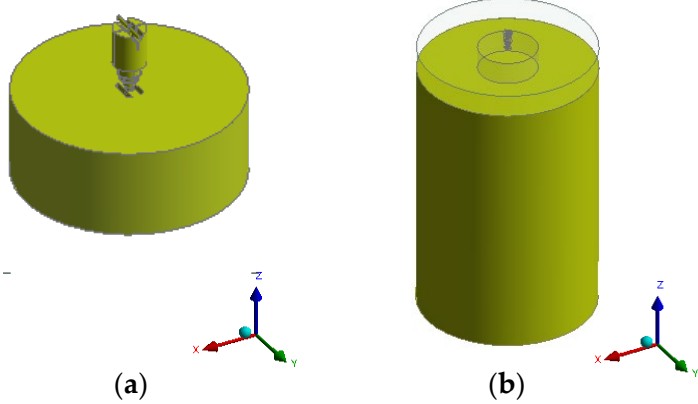

(**a**)    (**b**)

**Figure 7.** Calculation domain diagram: (**a**) Internal computing domain and (**b**) external computing domain.

(3)    Control Equation and model

In this paper, the fluid volume method (VOF) is used to simulate the air–liquid flow of AIN, which describes the conservation equation of volume fraction transport as follows:

$$\frac{d}{t}\int_V \varepsilon_i dV + \int_S \varepsilon_i(v - v_g)da = \int_V \left(S_{\varepsilon i} - \frac{\varepsilon_i}{\rho_i}\frac{D\rho_i}{Dt}\right) dv_i \tag{6}$$

where $V$ is total volume of poly phase flow; $v$ is mixing speed (m·s$^{-1}$); $a$ is the surface area vector; $\varepsilon_i$ is the phase volume fraction; $S_{\varepsilon i}$ is the Lagrangian derivative of phase $i$ density; and $D\rho_i/Dt$ is the Lagrangian derivative of phase $i$ mass fraction.

In the VOF method, the Navier–Stokes equation can be used to solve the equation for two fluids of different viscosity and density. Thus, the continuity and momentum equations for each phase are as follows:

$$\frac{\partial}{\partial t}\left(\int_V \rho dV\right) + \oint_A \rho v \cdot da = 0 \tag{7}$$

$$\frac{\partial}{\partial t}\left(\int_V \rho v dV\right) + \oint_A \rho v \otimes v \cdot da = - \oint_A \rho I \cdot da + \int_A T \cdot da$$

$$+ \int_V \rho g d + \int_V f_b dV - \sum_i \int_A \varepsilon_i \rho_i v_{d,i} \cdot \otimes v_{d,i} da \tag{8}$$

where $P$ is intensity of pressure (Pa); $I$ is unit tensor; $T$ is stress tensor; and $f_b$ is the gravity vector.

In order to better describe the flow of gas and liquid in the wall, the AIR is solved by using the Realizable k-ε turbulence model and Coupled Implication Solver based on the vortex and pressure gradient in the Venturi tube closer to the real Reynolds number. The iterative residue is less than 0.00001.

### 2.3.3. Grid Partitioning

The flow field of the inner and outer parts of AIN is divided by using an unstructured grid. The internal air–liquid two-phase calculation domain is $100 \times 300 \times 50$ mm, and the external nozzle atomization calculation domain is $300 \times 500$ mm. Key parts of the model, such as liquid import, gas import, spray exit, were mesh encrypted [23,24], totaling 1.6 million. As shown in Table 1, when the grid is 1–1.6 million, the data vary considerably. However, when the mesh count is 1.6 million to 2 million, the data difference is less than 5%.

**Table 1.** Grid independence test.

| Project | Grid Quantity | | | |
|---|---|---|---|---|
| | **A/1.3 Million** | **B/1.6 Million** | **C/2.0 Million** | **D/2.2 Million** |
| maximum speed/(m·s$^{-1}$) | 2.374 | 2.545 | 2.587 | 2.612 |
| outlet speed/(m·s$^{-1}$) | 2.133 | 2.424 | 2.473 | 2.494 |
| maximum pressure/(MPa) | 0.152 | 0.183 | 0.192 | 0.198 |

The grid is tested for independence. Venturi pipes are set as inlets, nozzles as free-flowing outlets, and air intakes as free-flowing inlets (Figure 8). To accurately predict the flow of the cross-wall boundary layer, the rest of the nozzle is treated as a wall with no slip boundary.

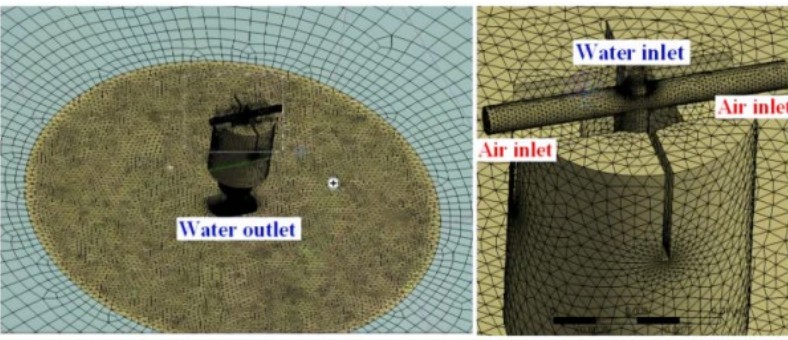

**Figure 8.** Grid encryption processing results.

## 3. Results

### 3.1. Analysis of Air–Liquid Flow Field Distribution

3.1.1. Velocity and Pressure Distribution Analysis

The velocity and pressure distribution of TD02-ST06 combined AIN along the centerline under a spray pressure of 0.6 MPa is shown in Figure 9. This paper divides the inside of the combined AIN into six regions according to speed and pressure distribution. The velocity is high in the pre-orifice region and then decreases in the throat region because of the increasing throat area. At the Venturi angle, as the area gradually increases, the velocity

decreases and the minimum velocity is 0.27 m/s in the air–liquid mixing chamber. At nozzle outlets, the velocity increases dramatically as pressure can be converted to kinetic energy. The change of pressure value in nozzle is the opposite of the change in velocity, which increases gradually as the area increases before the mixture reaches the nozzle. However, pressure increases first and then decreases in area C due to diffusion of mixtures at the Venturi angle.

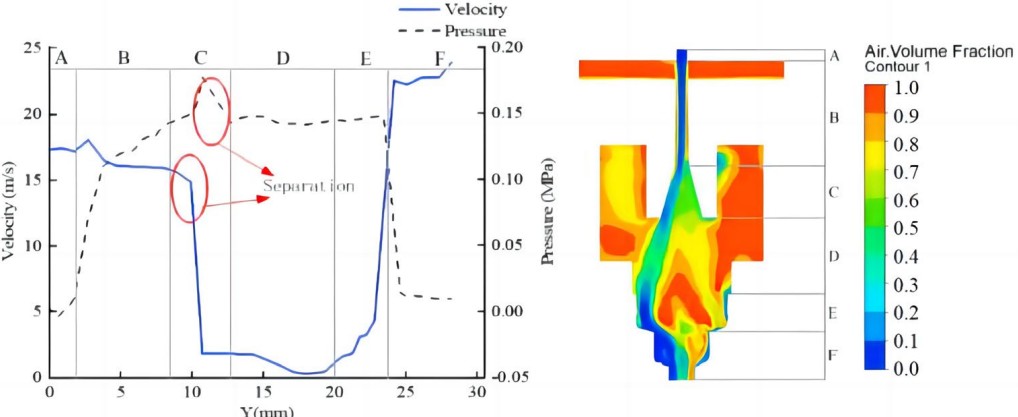

**Figure 9.** The velocity and pressure distribution along the centerline of AIN.

The results of Figure 9 show that when a liquid is ejected into the Venturi throat at a higher velocity, a low-pressure zone is created around the Venturi throat, causing air to be drawn into the throat. The pressure inside the nozzle is lower than the ambient pressure and acts as the driving force for entrained air. In the diffusion area of Venturi, as the cross-sectional area gradually increases, the velocity decreases gradually and the kinetic energy is converted into pressure energy.

### 3.1.2. Air–Liquid Volume Fraction Analysis

The volume distribution of the combined AIN in the mixing zone is shown in Table 2. With the increase of spray pressure, the air volume flow rate $M_a$, liquid volume flow $M_l$, and the ratio $M_a/M_l$ of each group of AIN increased gradually. The air volume flow rate of TD03-ST06 combined AIN is up to 0.0092 (L/min) at 0.6 MPa. However, under the same spray pressure, the volume flow of air does not increase continuously with the size of the Venturi pipe. This is because the flow of the combined AIN is determined only by the size of the Venturi pipe at the same spray pressure. Although a larger throat size can produce a larger volume flow of air, the volume flow of liquid also increases, resulting in a non-linear continuous increase in $M_a/M_l$. It can be seen from the air–liquid volume fractional distribution that the TD03 type Venturi tube can produce a larger air–liquid ratio when used in combination with the ST06 nozzle.

### 3.1.3. Air–Liquid Ratio Data Analysis

The air volume fraction of three combined AINs in the throat region under a spray pressure of 0.6 MPa is shown in Figure 10. Under the same spray pressure, TD03-ST06 combined AIN throat region has the highest gas volume fraction. As $dp/dt$ increases, the area occupied by the liquid jet gradually increases. However, as the diameter of the throat increases further, the flow rate of water decreases and the momentum of the entrained air decreases, causing the ALR to decrease. In addition, increased liquid flow decreases the volume fraction of the air volume, and this is consistent with the results shown in Table 2.

**Table 2.** Volume fraction cloud results of air in AIN.

| Nozzle Type | Pressure (MPa) | A-A' | $M_a$ (L/min) | $M_l$ (L/min) | $M_a/M_l$ |
|---|---|---|---|---|---|
| TD02-ST06 | 0.4 | | 0.0036 | 0.95 | 0.0038 |
| | 0.5 | | 0.0044 | 1.08 | 0.0041 |
| | 0.6 | | 0.0051 | 1.16 | 0.0044 |
| TD03-ST06 | 0.4 | | 0.0061 | 1.58 | 0.0039 |
| | 0.5 | | 0.0078 | 1.79 | 0.0044 |
| | 0.6 | | 0.0092 | 1.96 | 0.0047 |
| TD04-ST06 | 0.4 | | 0.0045 | 1.98 | 0.0022 |
| | 0.5 | | 0.0073 | 2.21 | 0.0033 |
| | 0.6 | | 0.0089 | 2.46 | 0.0036 |

*(Header: Volume Fraction of Air — scale 0.0, 0.3, 0.6, 1.0)*

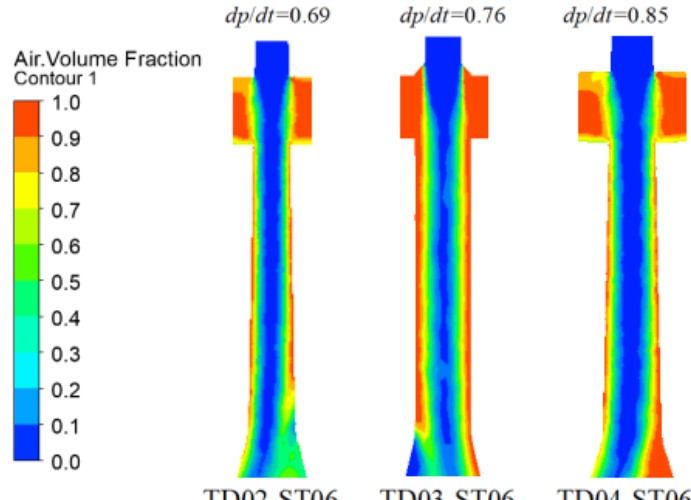

**Figure 10.** Contour of the air volume fraction at the throat region.

The instantaneous intake simulation and test monitoring data of TD03-ST06 combined AIN are shown in Figure 11. Before the test, due to the existence of air in both the nozzle and the flow meter, the volume flow of air was negative at the beginning of the monitoring stage. After iteration and test for a period of time, the simulation and test monitoring data show a stable trend. As the iteration time increases, the simulation and experimental data show a steady trend and remain at 0.002 L/min. It shows that the simulation analysis is consistent and accurate with the test results. As shown in Figure 12, the ALR of TD-ST

combined AIN under 0.3, 0.4, 0.5, and 0.6 Mpa is simulated and measured. Each group of tests was repeated three times to take the mean value.

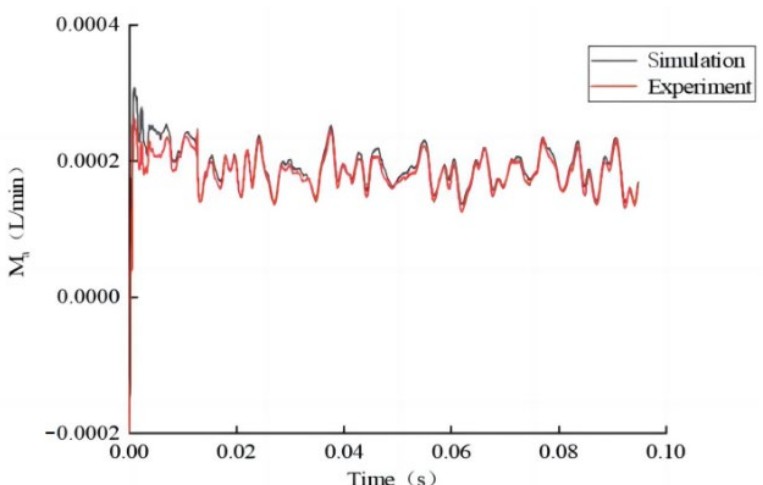

**Figure 11.** Droplets movement trajectory at different charging voltage.

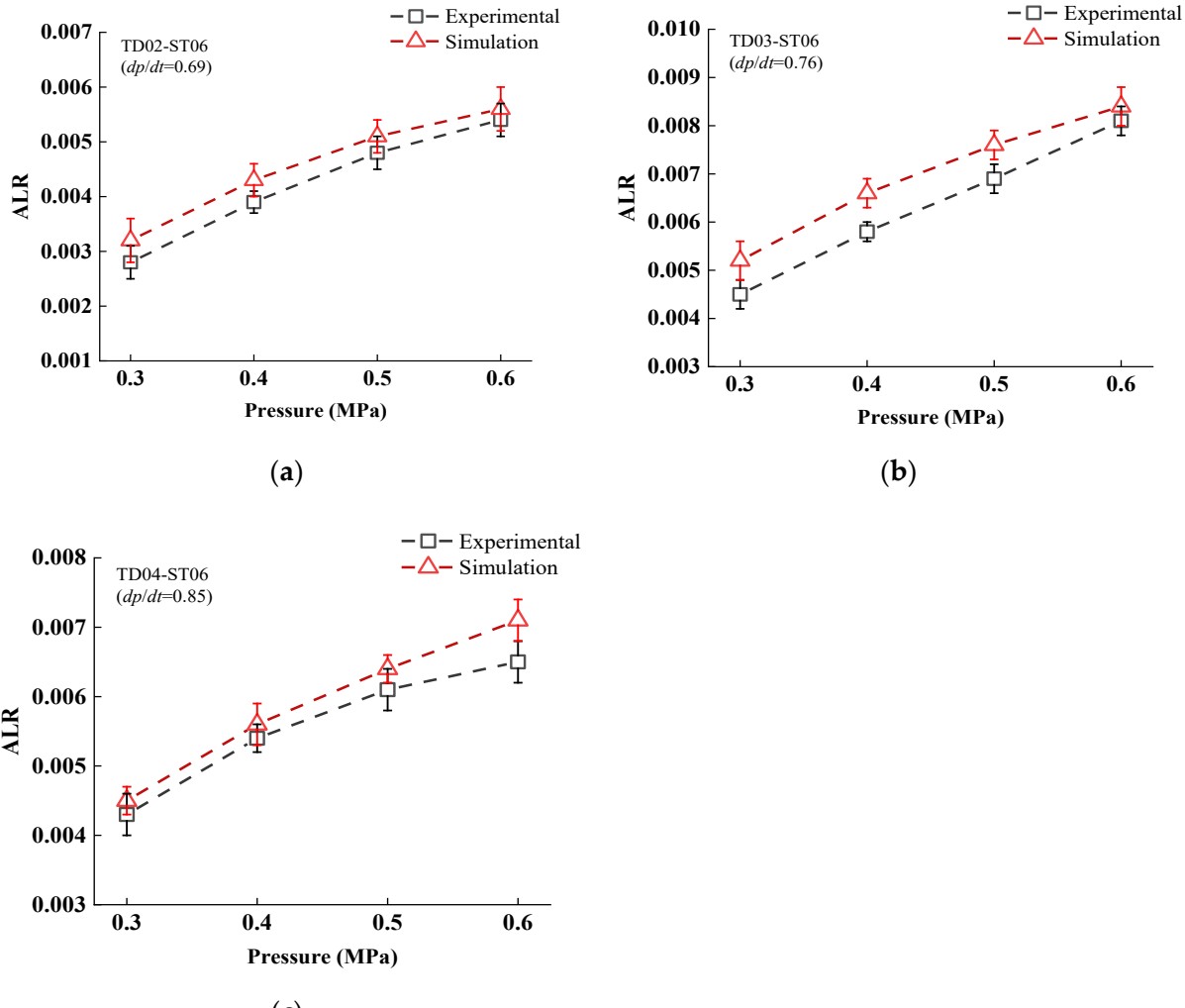

(**a**)

(**b**)

(**c**)

**Figure 12.** ALR of simulation and experiment: (**a**) TD02-ST06 combined AIN; (**b**) TD03-ST06 combined AIN; and (**c**) TD04-ST06 combined AIN.

As can be seen from Figure 12, the ALR of all three combinations increases with the spray pressure, and the simulation data of each group of ALR is higher than the test data. This is due to the loss of the pipeline of the sensor monitoring system along the route during the actual test, resulting in a portion of the gas not being monitored. When the spray pressure is 0.6 MPa and the TD-03 model Venturi tube is combined with the ST-06 type nozzle, the maximum value of simulation ALR is 0.0084 and test measurement ALR is 0.0081. When the $dp/dt$ increases gradually, the ALR increases first and then decreases. The result is consistent with the volume fraction distribution rule in Table 2.

### 3.2. Air–Liquid Ratio Regression Model Analysis

With spray pressure $P$, aperture ratio $K$ and nozzle slot width $V$ as independent variables and air–liquid ratio of TD-ST combined AIN as dependent variables, the test scheme and results are shown in Table 3. Based on the multi-linear fitting of the data in Table 3, the air–liquid ratio regression model of TD-ST combined AIN and the prediction results are shown in Figure 13. The coefficients of the fitting equation are shown in Table 3. The variance significance of the model was analyzed as shown in Table 4.

**Table 3.** Test protocols and results.

| Pressure (P)/MPa | Aperture Ratio (K) | Slit Width (V)/mm | ALR | Pressure (P)/MPa | Aperture Ratio (K) | Slit Width (V)/mm | ALR |
|---|---|---|---|---|---|---|---|
| | 0.69 | 0.99 | 0.0014 | | 0.69 | 0.99 | 0.0022 |
| | 0.69 | 1.20 | 0.0021 | | 0.69 | 1.20 | 0.0033 |
| | 0.69 | 1.40 | 0.0028 | | 0.69 | 1.40 | 0.0039 |
| | 0.76 | 1.20 | 0.0034 | | 0.76 | 1.20 | 0.0044 |
| 0.3 | 0.76 | 1.40 | 0.0045 | 0.4 | 0.76 | 1.40 | 0.0058 |
| | 0.76 | 1.80 | 0.0039 | | 0.76 | 1.80 | 0.0046 |
| | 0.85 | 1.40 | 0.0043 | | 0.85 | 1.40 | 0.0054 |
| | 0.85 | 1.80 | 0.0047 | | 0.85 | 1.80 | 0.0061 |
| | 0.69 | 0.99 | 0.0031 | | 0.69 | 0.99 | 0.0042 |
| | 0.69 | 1.20 | 0.0042 | | 0.69 | 1.20 | 0.0047 |
| | 0.69 | 1.40 | 0.0048 | | 0.69 | 1.40 | 0.0054 |
| 0.5 | 0.76 | 1.20 | 0.0056 | 0.6 | 0.76 | 1.20 | 0.0065 |
| | 0.76 | 1.40 | 0.0061 | | 0.76 | 1.40 | 0.0071 |
| | 0.76 | 1.80 | 0.0065 | | 0.76 | 1.80 | 0.0069 |
| | 0.85 | 1.40 | 0.0068 | | 0.85 | 1.40 | 0.0078 |
| | 0.85 | 1.80 | 0.0072 | | 0.85 | 1.80 | 0.0081 |

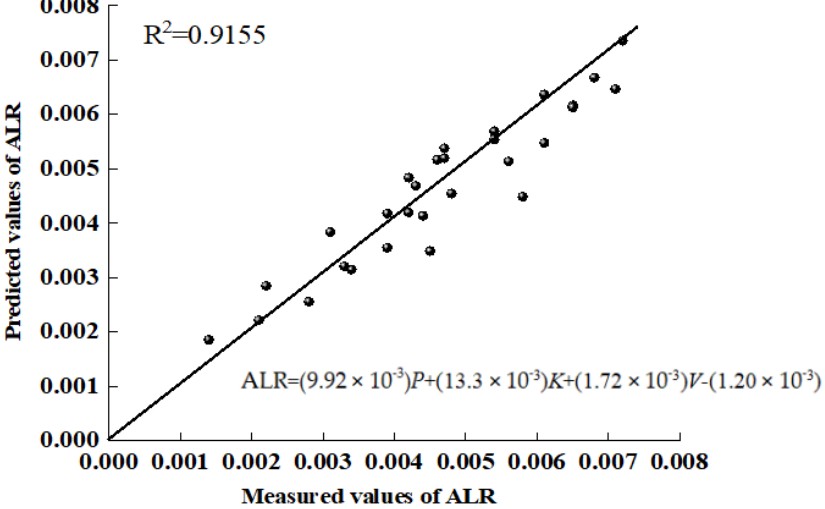

$$\mathrm{ALR}=(9.92 \times 10^{-3})P+(13.3 \times 10^{-3})K+(1.72 \times 10^{-3})V-(1.20 \times 10^{-3})$$

$R^2=0.9155$

**Figure 13.** TD-ST combined AIN air-liquid ratio regression model.

**Table 4.** Coefficient table of fitted equations.

| Factor | Unstandardized Coefficient | | Standardized Coefficient | Value T | 95% CI (Asymptotic) |
|---|---|---|---|---|---|
| | **Value B** | **Standard Error** | **Value Beta** | | |
| (Constant) | −0.001201 | −0.001208 | | 9.947 | −0.01449 to −0.009539 |
| Pressure (*P*) | 0.009924 | 0.000823 | −2.0688 | 12.05 | 0.008238 to 0.01161 |
| Aperture ratio (*K*) | 0.01333 | 0.001846 | 3.4095 | 7.219 | 0.009547 to 0.01711 |
| Slit width (*V*) | 0.001715 | 0.000429 | −4.7211 | 3.987 | 0.0008332 to 0.002594 |

The TD-ST combined AIN air-liquid ratio regression model formula is shown in (9).

$$ALR = (9.92 \times 10^{-3})P + (13.3 \times 10^{-3})K + (1.72 \times 10^{-3})V - (1.20 \times 10^{-3}) \tag{9}$$

where *P* is pressure (MPa); *K* is aperture ratio; and *V* is slit width (mm).

It can be seen from Tables 4 and 5 that the main effects on air–liquid ratio and spray pressure, aperture ratio, and slit width have significant influences on the air–liquid ratio model. Aperture ratio is the main factor that determines the air–liquid ratio of AIN, and the total ALR increases with the increase of aperture ratio. Spray pressure is a secondary factor affecting the ALR. With the increase of spray pressure, the ALR increases gradually. The slit width has little influence on the ALR. For the air–liquid ratio regression equation, the determinant $R^2 = 0.9155$ and the adjusting determinant $R^2_{adj} = 0.9065$ show that the model is credible.

**Table 5.** Significance analysis of variance.

| Type | Sum of Square | Degree of Freedom | Mean Square | Significance |
|---|---|---|---|---|
| Pressure (*P*) | $3.940 \times 10^{-5}$ | 1 | $3.940 \times 10^{-5}$ | $p < 0.0001$ |
| Aperture ratio (*K*) | $1.414 \times 10^{-5}$ | 1 | $1.414 \times 10^{-5}$ | $p < 0.0001$ |
| Slit width (*V*) | $4.312 \times 10^{-5}$ | 1 | $4.312 \times 10^{-5}$ | $p < 0.0001$ |
| Regression | $8.233 \times 10^{-5}$ | 3 | $2.744 \times 10^{-5}$ | $p < 0.0004$ |
| Residual | $7.597 \times 10^{-5}$ | 28 | $2.713 \times 10^{-5}$ | |
| Total | $8.992 \times 10^{-5}$ | 31 | | |

## 4. Discussion

During the use of standard fan nozzle and AIN, the researchers mainly studied the relationship between operation parameters and spraying quality. As researchers believe that the atomization effect plays a major role in pest control [14,19], the contrast atomization control effect is taken as a means to verify the superiority of spray parameters, and the research on the structural characteristics AIN are often neglected. Complex experimental conditions are required due to the small size of conventional AIN nozzles (such as IDK manufactured by LECHLER). Most of the research work on the characteristics of the ALR and spray of AIN remain in experimental studies by changing geometric shapes [21]. In order to grasp the internal flow characteristics of AIN, this paper uses a computational fluid dynamics method (CFD) to describe the process of air–liquid coupling model in detail. The results clearly reveal the air–liquid coupling phenomenon and the complete instantaneous turbulence characteristic occurring inside the nozzle.

The ALR is the key parameter in the design of an air induction nozzle since higher ALR contributes to larger sized droplets including micro-bubbles [12,15–17]. However, few similar studies have been conducted for the air induction nozzle covering the effect of the design parameters on the ALR, and they mostly focused on the droplet size measurements [18,19]. These studies have shown that identifying the influencing factors of ALR and stabilizing ascent based on operating conditions is a necessary scientific issue. It is also found that the ALR is influenced by the structure of the venturi tube and fan nozzle, similar to the findings of Vashahi et al. [20]. On the basis of previous research results, the specific monitoring methods of AIN intake volume are given. It was found that the ratio of inlet diameter to pipe diameter $dp/dt$ is the main design parameter affecting ALR. As

$dp/dt$ grows, the ALR tends to increase first and then decrease. Therefore, a deep grasp of the internal flow characteristics of AIN and the rule of ALR influence can achieve ideal spray effect.

## 5. Conclusions

In order to quantify the design parameters of the nozzle and the influence of spray pressure on the intake and air–liquid ratio of AIN, TD type Venturi tubes and ST type fan nozzles were combined to study the coupling law of air–liquid two-phase flow field. The research conclusions are as follows:

(1) The CFD numerical simulation method can be used to visually analyze the flow characteristics in AIN, such as velocity, pressure, air–liquid two-phase distribution, and inlet volume of the Venturi tube. The above parameters can well describe the transient flow behavior in AIN and clarify the air–liquid coupling law in AIN.

(2) The aperture ratio $dp/dt$ of combined AIN significantly affects the distribution of air–liquid two-phase fluid. When the spray pressure is 0.6 MPa and the $dp/dt$ is 0.76, (TD-03 model Venturi is combined with the ST 110-06 nozzle), the ALR reaches the maximum value of simulation and test measurement. The test results coincide with the simulation results.

(3) By means of multivariate linear regression, a model of pressure, aperture ratio, slit width of AIN, and AIR-liquid ratio regression is established. The variance analysis shows that the model has high significance ($p < 0.0001$, $R^2$ is 0.9155) and is suitable for the air–liquid ratio prediction of TD-ST combined AIN nozzle. This study provides theoretical basis and experimental guidance for the design and manufacturing of air induction nozzles with anti-drift properties.

**Author Contributions:** J.H. and C.L. conceived the idea of the experiment; Y.L. and C.L. performed the field test; Q.L. and C.L. analyzed the data; and J.H., C.L., Y.L., Q.L., S.Z. and W.Z. wrote and revised the paper. All authors have read and agreed to the published version of the manuscript.

**Funding:** This research was funded by Heilongjiang Province Applied Technology Research and Development Program Project (No. GA21B003).

**Data Availability Statement:** Not applicable.

**Conflicts of Interest:** The authors declare no conflict of interest.

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
