# Peer review of "Numerical Simulation on Air-Liquid Transient Flow and Regression Model on Air-Liquid Ratio of Air Induction Nozzle"

_agronomy, doi:10.3390/agronomy13010248_

Round 1

Reviewer 2 Report

1.     The abbreviations TD and ST should be explained in the sentences "TD type Venturi tubes" and "ST110 type fan nozzles" in the abstract.

2.     "TD-ST combined AIN" and other frequently used words should be included in key words.

3.     The measurement of “Line 149: “ca is the airflow velocity (kg·m−3);” airflow velocity” was given in kg·m−3, actually it should be in m/s.

4.     In pаrt 2.1, it is necessary to explain how TD02, TD03 and TD04 air flow nozzles differ from each other. Similarly, the difference of ST04, ST05, ST06 and ST08 air flow nozzles from each other should be noted.

5.     β is the void fraction coefficient” should be instead of “β is the void fraction” in Line 149.

6.     It is necessary to specify what ALR is after 2 or 3 formula in the 2.3 part.

7.     Line 168: “2.3.2. Computational fluid dynamics (CFD) numerical simulation method” should be written instead of “2.3.2. CFD numerical simulation method”, it will be more optimal.

8.     The TD-ST combined AIN air-liquid ratio regression model formula should be given in the text in the 3. Results section.

9.     The Discussion section should be written more widely. For example, the significance of the research results, the differences or similarities with the results obtained in other similar studies, etc.

Reviewer 3 Report

The manuscript entitled “Numerical simulation on air-liquid transient flow and regression model on air-liquid ratio of air induction nozzle” is interesting.

Some suggestions are as followed to improve the content.

1)      Line 36- Kindly include the citation for the first statement.

2)      Line 83- please check whether it should be “verification” or “validation”. These two terms serve two different meaning.

3)      Line 88- The abbreviation of CFD shall be described when you first mention it.

4)      Line 173- version of ANSYS shall be mentioned.

5)      Figure 7- remove the background colour “light blue”. Dimension shall be included (manual label instead of using ansys dimension function).

6)      Lin 188- Please check subscript of phase volume fraction.

7)      Line 193- Please check whether P is “intensity” of pressure.

8)      How do authors ensure the gid is sufficient for this study? Any grid independent test was performed? How about Grid convergence index? If no, kindly provide reasons to support your claims.

9)      Please revise figure 8 caption. Also, why tetrahedral mesh was used on the water inlet and air inlet, while hexahedral mesh was used on other zone. Kindly justify. Is it tetrahedral elements more suitable for complex shape, especially at the water inlet there? If yes, could cite this recent published article to support your selection.

https://doi.org/10.1016/j.buildenv.2022.109489

10)  Also, a blend of fine and coarse mesh was used in the study. Kindly provide justification too. Perhaps can mention a blend of fine mesh and coarse mesh is commonly used to reduce calculation time, yet provide reasonable results. Can consider citing the following article to support your statement.

https://doi.org/10.1007/s10973-022-11466-6

11)  Kindly enlarge Figure 11. Also, please provide a thorough discussion on the presented plot.

Reviewer 4 Report

The paper tried to numerically studies the effect of geometrical configuration of the air induction nozzle on the air-liquid ratio. The topic is okay as a reference in the related field. Please refer to the following comments to improve your quality of work.

Please include more discussion on why ALR can be used as the only criterion to measure the nozzle atomization. The references you cited in line 58 did not make such conclusions. Please explicit it, as this is the most important assumption to support the validity of your work.

Line 142: what does bubble-like two-phase flow mean? Probably the paper refers to bubbly two-phase flow.

Figure 7 shows the simulation performed in two domains. However, there is no results or discussion on the external computing domain. Why including it in your model? 

There is not enough information on the numerical setup, which prevent the readers from possibly replicating your results.

Did you perform any mesh sensitivity study? 

Please check the void fraction distribution results. Make sure the simulation results are converged. Are you simulating in transient or steady state mode? The latter one can rarely produce your kinds of void fraction distribution based on your description of CFD setup.

Round 2

Reviewer 1 Report

The authors have made most of the required changes.

Therefore, I again suggest to add the graphic scale of the colors with the corresponding values in fig.9 and enlarge fig.12 because it is not very clear.

Author Response

fig.9 and fig.12 have been modified

Reviewer 4 Report

Main issue 1

The author did not response directly to this question. 2nd version line62 "Previous researchers have not paid much attention to this": why didn't they pay attention to this parameter? Is it because this parameter is not as important as the droplet size in terms of measuring atomization characteristics? The fact is that this paper performed numerical study without focusing on the droplet size, but the ALR only. why not including droplet size into your analysis? The author should fully address this in your introduction.

Main issue 5: Did you perform any mesh sensitivity study?.

Response 5: Grid independence test is shown in line 208-210.

You should show detailed results in terms of your grid independence test, just like showing the simulation results. simply few words cannot prove the amount of work you should do in this part. This is an important content to convince reader the veracity of your results.
